# Bioactive Profile of Various *Salvia officinalis* L. Preparations

**DOI:** 10.3390/plants8030055

**Published:** 2019-03-06

**Authors:** Martina Jakovljević, Stela Jokić, Maja Molnar, Midhat Jašić, Jurislav Babić, Huska Jukić, Ines Banjari

**Affiliations:** 1Faculty of Food Technology, Josip Juraj Strossmayer University of Osijek, Franje Kuhaca 20, 31000 Osijek, Croatia; mjakovljevic@ptfos.hr (M.J.); mmolnar@ptfos.hr (M.M.); jbabic@ptfos.hr (J.B.); ibanjari@ptfos.hr (I.B.); 2Faculty of Technology Tuzla, University of Tuzla, Univerzitetska 8, 75000 Tuzla, Bosnia and Herzegovina; jasic_midhat@yahoo.com; 3Department sanitary engineering, University of Bihac, Faculty of Health Studies, Nositelja hrvatskog trolista 4, 77000 Bihać, Bosnia and Herzegovina; huskaj037@gmail.com

**Keywords:** *Salvia officinalis* L., sage preparations, extraction techniques, bioactive compounds, health effects

## Abstract

*Salvia officinalis* L., also known as the “Salvation Plant”, has been long used and well-documented in traditional medicine around the globe. Its bioactive compounds, and especially its polyphenol profile, have been extensively researched and reviewed. However, sage’s beneficial effects reach much further, and nowadays, with a range of new extraction techniques, we are discovering new components with new therapeutic effects, especially in the context of neurodegenerative diseases and various carcinomas. This review describes the bioactive profile of various sage preparations depending on the extraction techniques and extraction parameters, and this review lists the newest research findings on its health effects.

## 1. Introduction

*Salvia officinalis* L. is a plant in the mint family Lamiaceae, subfamily Nepetoideae, tribe Mentheae, and genus Salvia [1]. Salvia is the largest genus of the Lamiaceae family, containing around 1000 species [2], and can be found in Europe around the Mediterranean, in Southeast Asia, and Central and South America [3].

*S. officinalis* grows in the form of an outcrossing, perennial subshrub up to 60 cm high. The leaves are opposite and simple with white hairs on the lower leaf surface and greenish or greenish-grey on the upper surface. Stems are erect or procumbent with abundant hairy dark green branches. Leaves are elongated and petiolate with a serrate margin, rugose surface, and sometimes with basal lobes. The flowers are 2–4 mm long from the pedicel, and they are in pseudoverticillasters with 5–10 violet-blue color flowers that form spurious, composed spikes. They bloom from March to July depending on habitat and climatic conditions [4,5].

Historically, sage is known as the “Salvation Plant”, originating from the old Latin word “salvarem”, which means save or cure. It has been used to reduce perspiration, as a gargle for sore throat, to improve regularity of a menstrual cycle and to reduce hot flashes in menopause, to fight gastroenteritis and other infections, to improve lipid status and liver function in general, to improve appetite and digestion, and to improve mental capacity [6]. Recently, focus has been put on the link between specific bioactive compounds in sage and specific health effects. Additionally, more effort has been put in to determine the best extraction method, conditions, and parts of a plant to be used in order to get the most effective preparation.

This is the first review to encompass the extraction techniques, composition, and health effects of various sage preparations. Previous review articles covered the topic of pharmacological properties and components [7,8], biochemical studies [9], medical properties, and genetic diversity of Dalmatian sage [10], as well as chemistry, pharmacology and medicinal properties for the prevention and treatment of various health conditions [11], chemistry and antioxidative factors in sage [12], and polyphenolics of Salvia [13].

## 2. Methodology of Review

Hereby, we used scientific databases, such as PubMed, ScienceDirect, Scopus, ResearchGate, and Google Scholar, with emphasis on ScienceDirect and Scopus. Review of the literature was carried out using the keywords “Sage”, “Salvia officinalis”, and “extraction”. According to these key words we obtained 511 references in ScienceDirect and 101 references in Scopus during February and March 2017. During the search of literature and writing of the review, the time period in which the papers were published had not been selected, since the focus was on significant works selected for the areas covered in this review. All data were analyzed in corresponding articles.

## 3. Production of Sage Extracts

Today, a number of various techniques are used for the production of various sage products. The techniques are chosen depending on the desired profile of sage’s bioactive compounds in an extract.

### 3.1. Hydrodistillation

There are several processes for obtaining sage products mentioned in the scientific literature. Most of the research involving this plant is focused on the production of essential oil and its chemical composition. The most commonly used methods are conventional processes including hydrodistillation, using a Clevenger-type apparatus for 30 min [14], 1 h [14], 2 h [14,15,16,17,18,19], 3 h [20,21,22,23,24,25,26] or 4 h [27,28,29,30,31], the modified Clevenger apparatus for 2 h [32,33], and an Unger-type apparatus for 3 h [34]. The hydrodistillation apparatus can be placed inside a microwave oven to obtain essential oil without any addition of solvents, including water, as described by Koubaa et al. [35]. Hydrodistillation was performed using a microwave power of 500 W and a temperature of 100 °C for 30 min.

### 3.2. Soxhlet Extraction

Certain sage extracts were obtained by Soxhlet extraction with different solvents, as well as the time of the extraction [29,36,37]. In Farhat et al. [36], methanol was used as the extraction solvent, and the extraction was performed for 2 h, while in the case of Kontogianni et al. [37], two solvents of different polarity, hexane and ethyl acetate, were used for an extraction that lasted for 6 h. Soxhlet extraction of sage leaves was also conducted by using mixture of ethanol and water (70:30 *v*/*v*) for 4 h [29].

### 3.3. Infusion

Infusion of sage leaves, or so-called production of sage tea, is a very popular preparation in folk medicine. This process is very simple, but is conducted quite differently in the research. According to Radulescu, Chiliment, and Oprea [38], 100 mL of boiling water are poured over 5 g of leaves of *Salvia officinalis* L. and filtered after 30 min. In Martins et al. [39], 200 mL of boiling water was poured over 1 g of sample, left for 5 min, and then filtered under reduced pressure. In Zimmermann et al. [40], 150 mL of boiling water was poured over 1.5 g of sage leaves or tea bags from 16 different brands, steeped for 15 min, and then a filtered 1 mL sample was used for further analysis.

### 3.4. Solid–Liquid Extraction

In addition to hydrodistillation, a significant number of studies of *Salvia officinalis* L. were conducted on the products obtained by solid–liquid extraction by using different solvents and comparing both classical and innovative extraction techniques. 

Dent et al. [41] used three different aqueous solutions of ethanol (30%, 50%, or 70%), acetone (30%, 50%, or 70%) and distilled water to extract polyphenols, which were determined by the Folin-Ciocalteu method and the HPLC UV/PDA (High-Performance Liquid Chromatography Ultraviolet/photodiode array) method. In addition to various solvents, the extraction took place at different temperatures (60 and 90 °C) for different times (30, 60, and 90 min) to show whether these parameters influenced the amount of total and individual polyphenols. The results showed that extraction with an aqueous solution of ethanol or acetone (30%) at 60 °C for 30 min was the most effective method for extracting polyphenols from dried sage leaves.

The influence of temperature, extraction time, solvent composition, sage particle size, and solvent-to-sage ratio was examined by Durling et al. [42]. The authors studied the efficacy of extraction of carnosic and rosemary acid as well as the yield and composition of sage essential oil. The optimum conditions for the highest yield of carnosic compounds, rosemary acid, and essential oil, which were 10.6%, 6.9%, and 7.3% respectively, were a 1 mm sample size of dried sage, 55–75% ethanol, and a solvent-to-sage ratio of 6:1 at 40 °C for 3 h. It is well-known that diffusion of the solvent is better when the plant material is ground to the smallest particle size, but grinding can cause dust generation and heat production, which may affect the composition of the plant material itself. The increase in temperature increases yield, but at a certain temperature, in this case up to 40 °C, it can cause a reduction in the yield and the evaporation of the volatile components. By increasing the time of extraction, yields and total polyphenols did not increase, i.e., they were constant, but the yield of carnosic components, rosemary acid, and essential oil increased, with the conclusion that other components were unstable for a longer duration of extraction. Therefore, the authors recommended the duration of the extraction for no longer than 3 h. As the extraction of the bioactive components was limited by the solubility in the solvent mixture used, it is important to find a suitable solvent ratio and solvent-to-sage ratio to reduce the cost of solvents and the energy associated with solvent evaporation.

Duletić-Laušević et al. [43] extracted the sage material with dichloromethane (DCM), chloroform, ethyl acetate, and ethanol for 24 h at 30 °C before and after the ultrasound treatment for 1 h. According to their results, yield was higher in plants that originated from Serbia (2.50%) than from those in Montenegro (2.03%), and yield was higher in plants that were extracted with dichloromethane (3.23%). Harvest season did not influence the yields. The content of total polyphenols and flavonoids depended on the extraction solvent and harvest season, and was higher in those plants originally from Serbia and harvested in the summer. When ethanol was used as a solvent it exhibited the highest influence on content of polyphenols, unlike ethyl acetate, which had the highest influence on content of flavonoids. Extracts of sage harvested in the summer and extracted with ethanol showed a better antioxidant activity, proving correlation between total polyphenols and antioxidant activity, which confirms the fact that polyphenols have more effect on antioxidant activity than flavonoids.

In the research by Roby et al. [44], sage extracts were prepared with solvents of different polarity (methanol, ethanol, diethyl ether, and hexane) with shaking at room temperature for 72 h. The yield differed depending on the solvent used, and the highest yield of 23.41% ± 2.65% was achieved with methanol, whereas the lowest yield (4.63% ± 1.73%) was obtained with hexane. Likewise, the highest number of total polyphenols was found in the extracts with methanol and ethanol (5.95% ± 2.65%; 5.80% ± 1.00%), while the lowest amount was found in hexane samples (4.25% ± 1.00%). These results were expected since the polar solvents are more effective in extracting phenolic components than less polar solvents.

#### 3.4.1. Ultrasound-Assisted Extraction (UAE)

Application of UAE is expanding, so Sališová, Toma, and Mason [45] compared conventional and ultrasound assisted extraction in the content of active components including cineole, thujone, and borneol using 65% ethanol as solvent. They studied the effect of temperature, stirring, and ultrasound (ultrasonic bath or horn system) for 12 h, and the concentration of the components was measured not only during this 12 h period, but also after 7 days. The results showed that ultrasonic extraction for 12 h at room temperature with stirring had better results compared with convectional techniques. Stirring is an important factor since it was observed that almost the same results were obtained at 30 °C without stirring and at a temperature of 20 °C with stirring. Even better results were obtained with an ultrasound horn, where a 2 h extraction resulted in the same amount of bioactive components as a 12 h extraction, but the main disadvantage of this method is the inability to control the temperature.

Veličković et al. [46] examined the effect of ultrasound and classical maceration on the extraction yield and composition by selection of suitable solvent. UAE was performed on an ultrasonic bath for 20 min and 40 °C, while maceration was carried out for a period of 6 h at 20 °C, with petroleum ether, 70% ethanol, and water as a solvent. Among these solvents, a 70% solution of ethanol appeared to be the most appropriate because the yield had the largest number of typical components. As far as the yields were concerned, a dependence on the solvent polarity was observed, where the yields increased with the solvent polarity. Therefore, the greatest yield was in the case of water and ethanol, respectively.

#### 3.4.2. Microwave-Assisted Extraction (MAE)

MAE is also one of the innovative extraction techniques that have increasingly been used for plant extraction, including sage [35,47,48]. Dragović-Uzelac et al. [47] examined the influence of solvent (distilled water, 30% aqueous ethanol, and 30% aqueous acetone), extraction time (3, 5, 7, 9 and 11 min), as well as microwave power (500, 600, and 700 W) at a constant temperature of 80°C on extraction of total polyphenols from sage. They showed that ethanol and acetone were better solvents for the extraction of polyphenols from sage than water, especially when using a microwave power of 500 W for 9 min. In a paper by Putnik et al. [48], the same solvents were used, almost the same extraction times (3, 5, 7, 9, and 10 min), and 100–500 W microwave power, but the difference from the previous paper was in the optimization of extraction temperature (30, 50, 60, and 80 °C) and addition of 10% HCl for better hydrolysis of polyphenol conjugates. Their results are similar to those of Dragović-Uzelac et al. [47]; they showed that the total polyphenols were better extracted with 30% aqueous acetone at 80 °C for 10 min without HCl. Looking at individual polyphenols, the highest content of most common polyphenolic acids was extracted using 30% aqueous ethanol with the addition of HCl at 80 °C.

### 3.5. Supercritical CO_2_ Extraction (SC-CO_2_)

In addition to the above mentioned classical procedures, the scientific literature examines innovative extraction techniques that were developed to overcome the deficiencies of classical techniques. SC-CO_2_ extraction is one of the promising alternative technologies, characterized by a good dissolution ability equivalent to organic solvents with better diffusion, lower viscosity and lower surface tension of fluid, fast and easy separation of extract and solvent, easy removal and possibility to recycle supercritical fluid from extract, and extraction of thermolabile components with minimal deformations [49,50]. CO_2_ has proven to be a very desirable solvent due to the fact that it possesses convenient critical properties. It is natural, cheap and widely available, non-toxic, non-flammable, chemically inert, easily removable from the product, and has no taste nor smell [51]. By using SC-CO_2_ at increased pressure, separation of components that are less volatile, higher in molecular weight, and more polar was obtained. The highest affinity of SC-CO_2_ was achieved in the case of oxygenated organic components with medium molecular weight. Due to the nature of CO_2_, this method was excellent for extraction of vegetable oils with a very high nutritional value [52]. On the other hand, the lack of this type of extraction lies in the fact that CO_2_ is an appropriate solvent for a non-polar and slightly polar components, with low affinity for the polar components, which can be improved by using polar co-solvents (ethanol, methanol) [49,52,53,54].

Reverchon, Taddeo, and Porta [55] studied the influence of different SC-CO_2_ extraction process parameters on the composition of the sage extracts, compared to the essential oil obtained by hydrodistillation. The influence of different process parameters of SC-CO_2_ extraction on the composition of the sage extracts was examined in the 80–100 MPa pressure range at two temperatures 45–60 °C at a constant flow rate of 0.95 kg h^−1^. The results showed the optimum extraction conditions for the maximum percentage of oxygenated compounds were at 90 MPa and 50 °C. Process parameters that affected the composition of supercritical fluid sage extract the most were extraction time, solvent density, and pressure, since the essential oils are soluble at lower pressures. It has also been shown that the content of sesquiterpenes and diterpenes were higher in the extract obtained by supercritical fluid extraction, depending on the parameters used then in essential oil obtained by hydrodistillation.

In another study by Daukšas et al. [56], the influence of different process parameters of SC-CO_2_ extraction was investigated in three separators (20, 25, and 30 MPa in the first separator, 10 MPa in the second separator, and 5 MPa in the third separator) to obtain the essential oil and to separate chlorophyll and waxes. This was performed at constant temperature of 40 °C and flow rate of 0.05 kg min^−1^ with 1% or 2% of ethanol as an extraction entrainer. They have shown that supercritical fluid extraction is effective for obtaining a pure extract, especially at 35 MPa and with the addition of 1% ethanol, whose addition increases the extraction yields. Additionally, antioxidant activity of these extracts was tested, indicating the dependence on the fractionation conditions. The stronger antioxidative effect is shown for the fractions extracted in the second and third separator. 

Fornari et al. [57] performed extraction and fractionation of sage at 30 MPa, 40 °C, and 2.4 kg h^−1^ of CO_2_ in two separators, working at 10 MPa and ambient pressure. The yield was measured in the period of 1.5 to 4.5 h, whereby the total yield after 4.5 h in separator 1 and 2 was 1.38% and 3.23%, respectively.

Not only Daukšas et al. [56] investigated the influence of ethanol as a co-solvent, Menaker et al. [21] also examined the influence of pressure and co-solvent on the yield and composition of the extract. The extraction was carried out for 1 h at pressures of 17.2 to 25.5 MPa, a temperature of 45 °C and a CO_2_ flow rate of 1 mL min^−1^. In this range it was observed that the pressure and the use of co-solvents affected the yield and the composition of the extract. It was also noticed that the increase in pressure and the co-solvent did not always have a positive effect on the yield, since with a 17.2 to 25.5 MPa pressure increase the concentration of 1,8-cineol and camphor increased, and the concentration of all of the other mentioned components decreased. 

In the case of supercritical fluid extraction, Fellah et al. [58] showed that the yield depended on the time and pressure, with the maximum yield at 160 MPa and 150 min. In addition, they showed that the total yield of essential oil produced by hydrodistillation for 300 min was equal to the yield obtained by the supercritical fluid extraction at 120 MPa and 270 min. 

Aleksovski and Sovová [34] determined a visual difference between the sage oil obtained by hydrodistillation and the extract obtained by supercritical fluid extraction. The essential oil was a transparent liquid of light-yellow color with specific odor, while supercritical fluid extracts were a light to dark yellow color with an aroma similar to the aroma of the plants themselves. The yield of the hydrodistillation process was 27 mL kg^−1^. In the case of supercritical fluid extraction, the yield ranged from 2.7% to 4.8% depending on the process parameters (9 to 12.8 MPa, 25–50 °C, and CO_2_ flow rate of 0.35 g min^−1^ in a time of 3 h). Maximum extraction yield of 4.8% was obtained at 12.8 MPa and 50 °C, proving that the extraction yield increased with both extraction temperature and solvent density. 

Glisic et al. [29] examined the influence of the process parameters on supercritical fluid extraction and the difference in the chemical compositions between the essential oil obtained by hydrodistillation and the extract obtained by the supercritical fluid extraction. They also compared their extracts with the one obtained by Soxhlet extraction using ethanol-water (70:30) mixture. The supercritical fluid extraction was carried out at a pressure of 7 MPa and 10–30 MPa at the temperature of 50 °C and a CO_2_ flow rate of 0.4 kg h^−1^. The yields differed depending on the technique used. The yield was 4.82% in the case of supercritical fluid extraction at the highest pressure of 30 MPa, the yield was 26.5% in the Soxhlet extraction, and the yield was 0.5% in the in process of hydrodistillation. They also examined the visual difference between the obtained extracts, which were in accordance with the investigation of Aleksovski and Sovová [34]. Their conclusion was that the supercritical fluid extract was yellow-brownish, the Soxhlet extract was dark green, and both were semi-solid, as opposed to the light-yellow liquid essential oil that was obtained by the hydrodistillation process.

At similar pressures as in Glisic et al. [29], Mičić et al. [59] extracted wild sage using the supercritical fluid extraction. The process parameters for supercritical fluid extraction were a pressure at 80 MPa and 100–300 MPa, a temperature at 40 °C, and a CO_2_ flow rate of 3.23 × 10^−3^ kg min^−1^ for 4 h. The extract yield and composition depended on the parameters used, with the highest yield of 4.65% at 30 MPa, which was in alignment with the results of Glisic et al. [29]. Isolation of the essential oil from the supercritical fluid extract showed that the highest proportion of essential oil was found in the extract at 80 MPa (59.79%). Occhipinti et al. [33] compared the extraction obtained by supercritical fluid extraction and the essential oil obtained by hydrodistillation to determine a more suitable method for α- and β-thujone extraction. Supercritical fluid extraction was carried out at pressure of 25 MPa, a temperature of 60 °C, a CO_2_ flow of 6 kg h^−1^ for 90 min, and a 2% ethanol co-solvent. The extracts were fractionated using two separators to separate the terpene fraction from heavy components. Considering the yields, the statistical analysis did not show any significant difference. The yield of essential oil was 90 ± 7.5 mg kg^−1^ and was 97 ± 3.7 mg kg^−1^ in supercritical fluid extract of the leaf dry weight. However, a noticeable higher concentration of α-thujone was found in the essential oil and the same amount of β-thujone in both extracts; therefore, the authors suggest that the production of essential oil by hydrodistillation was more cost-effective and economically more acceptable. Jokić et al. [60] investigated the influence of process parameters (pressure, temperature, and CO_2_ flow) on the yield and composition of the extracts, with an emphasis on oxygenated monoterpenes, α-humulene, viridiflorol, and manool. The SC-CO_2_ extraction was carried out at a pressure range of 10–30 MPa, a temperature of 40–60 °C, and a CO_2_ flow of 1–3 kg h^−1^ with a constant time (90 min) and particle size. The color of all the extracts was yellow-brown, and it was semi-solid with a strong aroma like the plant itself, as demonstrated by Aleksovski and Sovová [34] and Glisic et al. [29]. Pressure had a statistically significant effect on the yield and composition of extracts, while the temperature and CO_2_ flow did not exhibit significant influence. Thus, the optimum conditions for extraction of desired components were 13 MPa, a temperature 40 °C, and a CO_2_ flow rate of 3 kg h^−1^. 

In order to obtain the extracts with a higher concentration of desired components or antioxidative activity, Glisic, Ristic, and Skala [61] studied the ultrasound-assisted extraction with different solvents (water-ethanol mixture or just water), followed by re-extraction of obtained extract with SC-CO_2_ (150 MPa, 50 °C at flow rate of 0.4 kg h^−1^). The investigated SC-CO_2_ extraction conditions influenced the selectivity of the terpenes responsible for the antioxidant activity of sage. The best extraction procedure suggested was the ultrasound pretreatment of plant material with distilled water and re-extraction of plant material (residue) using SC-CO_2_ at 50 °C and 150 MPa.

## 4. Analysis of Different Extraction Methods

### 4.1. Hydrodistillation/Steamdistillation

Hydrodistillation is one of the oldest and most common techniques for essential oils extraction, mostly because of its simplicity. But at the industry level, steam distillation is used more often for the production of essential oil. Due to the heating and cooling in the process, especially at the industrial level, there is a high energy consumption and, therefore, higher extraction costs. These costs could be reduced by working under moderate pressures, because of the shorter distillation time, which reduces steam consumption or heat recuperation produced during vapor condensation, thus leading to energy savings without affecting the quality of the product itself [62].

### 4.2. Soxhlet Extraction

Soxhlet extraction is one of the frequent conventional methods used where the greatest drawbacks are long extraction times and use of hazardous and flammable organic solvents [63]. However, in this extraction method, less time and a smaller consumption of solvent is needed, compared to other conventional methods such as maceration or percolation [64].

### 4.3. Infusions

Infusion is one of the most general techniques used for galenical preparations, especially in the past when there was no industrial production of herbal extracts in large scales. Infusions are prepared by macerating the crude part of the plant with boiling water for a short period of time. These preparations are intended for consumption and considered freshly prepared. They are prepared in water and rapidly produce a deposit as a result of the coagulation of inert colloidal material. They also support microbial growth, and because of the fungal and bacterial growth, preparations must be consumed in a period of 12 h, which makes it unacceptable for industrial production. A period of a safe consumption can be extended with the addition of 25% alcohol during or after the extraction process, followed by dilution, according to the instructions of the European Pharmacopoeia [65]. 

### 4.4. Solid–Liquid Extraction

Maceration, as one of the most common forms of solid-liquid extraction, is a very simple extraction technique whose drawbacks are a long extraction time and low extraction efficiency [64]. Traditional maceration involves placing milled or crushed plant material in a closed vessel with the addition of a suitable solvent, with occasional shaking. In the case of industrial production, certain adjustments are required in order to allow better extraction efficiency, with as little solvent and evaporation costs as possible. Therefore, in the industrial production of extracts, the methods of constantly circulating solvents through plant material and a multistage extraction are used. 

Ultrasound-assisted extraction is a cost-effective and efficient technique compared to conventional extraction techniques. Compared to other alternative extraction techniques, equipment costs are lower, making this extraction technique suitable for industrial applications [63]. However, some authors claim that large-scale application is limited because of the higher costs [65].

Microwave-assisted extraction is one of the simple modern extraction techniques using low-cost equipment. Because of the microwave exposure it is possible to reduce extraction time and solvent usage. In processing applications, the possibility to shut the heat source affects product quality and production economics as well. Also, due to the possibility of processing a high volume of raw material over the same period, the time needed to get the final products is reduced. The problem arises in cases of the extraction of volatile and thermo-sensitive components, since cooling or venting periods are required after the extraction process [66].

### 4.5. Supercritical CO_2_ Extraction

Supercritical CO_2_ extraction is an alternative technology with numerous advantages, listed in Section 3.5, but it is limited by high investments, capital, and operating costs. With regard to high work pressure, heating, and cooling, the equipment and the process of extraction are expensive [67]. However, in the paper by Pereira and Meireles [68], cost of production of the SFE (Supercritical Fluid Extraction) extracts was lower than those produced by conventional methods. According to a paper by Shariaty-Niassar et al. [69], manufacturing costs (70%–85%) are the most expensive because of the use of high-pressure valves and pumps, especially at higher pressures and temperatures. This cost falls under fixed costs, while raw material, labor, CO_2_ supply, and utility are the operating costs. The usual costs for the SFE unit include the electricity required to operate the pump, a condenser, heating the water to warm up device, and cooling the water. In another paper [70] variable costs including costs connected with incomplete product recovery, CO_2_ pressing and pumping, cooling system, and idle time cost are calculated as production costs. One of the proposed approaches in reducing production costs is to maintain the variable stream circulation of the solvent through the extraction bed during a particular process step.

## 5. Chemical Composition of Sage Products

Sage contains many biologically active compounds that can be divided into monoterpenes, diterpenes, triterpenes, and phenolic components, given in Figure 1 and Figure 2.

Highly abundant phenolic components can be divided into two groups: phenolic acids (caffeic, vanillic, ferulic, and rosmarinic acids) and flavonoids (luteolin, apigenin, and quercetin) [13,44]. The most common monoterpenes include: α- and *β*-thujone, 1,8-cineole, and camphor. The most common diterpenes include: carnosic acid, carnosol, rosmadial, and manool. Triterpenes include oleanolic and ursolic acids [58,60,61,62,63]. In addition, sesquiterpene α-humulene and viridiflorol are also present in sage and extracts [23,29,60].

As previously mentioned, the bioactive composition of the sage product will depend on the extraction technique used, but also on the part of the plant used. Table 1 shows the bioactive composition of sage extracts obtained with hydrodistillation, depending on the part of the plant used, while Table 2 shows the bioactive profile of extracts obtained with supercritical fluid extraction. Bioactive profile for all other types of sage extracts are shown in Table 3. Extracts from sage leaves had the highest content of polyphenols, followed by aerial parts.

Most of the papers examine sage essential oils, representing the most investigated product of this plant, reporting different yields and compositions in various works (Table 1).

Essential oil is a secondary metabolite, whose production depends on conditions such as individual plant chemotypes, geographical location, date of harvest and harvest frequency, growing conditions, water deficit, proportions of plant parts and type of drying, as well as hydrodistillation time [14,17,18,20,23,24,26,27,71,72,73,74,75,76,77,78,79,80]. Given the difference in composition of essential oil shown in Table 1, some authors divide sage into several chemotypes depending on the concentration of the components. The most commonly used method is proposed by Tucker and Maciarello [81]. They have divided *Salvia officinalis* essential oils into five groups based on four main components in the following order: (1) camphor > α-thujone > 1,8-cineole > β-thujone; (2) camphor > α-thujone > β-thujone > 1,8-cineole; (3) β-thujone > camphor >1,8-cineole > α-thujone; (4) 1,8-cineole > camphor > α-thujone > β-thujone; and (5) α-thujone > camphor > β-thujone > 1, 8 cineole [81]. However, components like α-humulene, viridiflorol, or manool are highly abundant in sage essential oil. Therefore, Jug-Dujaković et al. [82] carried out a hierarchical cluster analysis based on the most common components (α-thujone, camphor, β-thujone, 1,8-cineole, β-pinene, camphene, borneol, and bornyl acetate) on samples from Croatia, in the Dalmatian region. Three clusters were created: (A) α-thujone > camphor > 1,8-cineole > β-thujone were able to be separated; (B) β-thujone > α-thujone > camphor ≈ 1,8-cineole; and (C) camphor > α-thujone > 1,8-cineole > camphene ≈ borneol. Hierarchical cluster analysis was also performed by Lakušić et al. [18] who showed that the chemotype also depended on the age of the leaves. The young leaves belonged to α-humulene chemotypes, while the old leaves belonged to camphor or a thujone chemotype depending on the country of origin (Serbia or Croatia). Craft, Satyal, and Setzer [83], based on a cluster analysis of volatile components, divided the essential oil of sage leaves into 5 chemotypes. The first of them, as the authors have called it C1, is α-thujone/camphor chemotype, which is divided into 3 subgroups that are identical to those of the aforementioned authors. The third subgroup, C1c, has the best composition of 8.0% α-thujone, 18.6% camphor, 10.5% 1.8-cineole, and 6.4% β-thujone. The second group, called C2, is an α-humulene-rich group that is also divided into three subgroups. The third group, C3, is a β-thujone-rich chemotype that is divided into two subgroups. The fourth group, C4, known as 1,8-cineole/camphor chemotype, is also divided into two subgroups. The fifth chemotype (C5) is a sclareol/α-thujone type. Perry et al. [75] found three chemotypes that differ in the content of α- and *β*-thujone, and, hence, divided sage into high (39%–44%), medium (22%–28%), and low (9%) content of thujone. Also, they showed that there was a difference between the flowering parts of sage and leaves. The flower parts contained a higher proportion of β-pinene (27% versus 10%) and a smaller of thujone (16% versus 31%).

Regarding Dalmatian sage, there is also ISO (International Organization for Standards) standard 9909: 1997 that refers to the required composition of essential oil.

As for other sage production processes, apart from harvesting conditions, geographical areas, and the plant itself, the yield and composition of the extract is influenced by the process parameters.

As mentioned above, in the case of SC-CO_2_ extraction, parameters such as the applied pressure, temperature, CO_2_ flow, time, co-solvent addition, etc., are those that affect the chemical composition [21,29,33,34,55,56,57,58,59,60,61,84]. Variable bioactive profiles of sage extracts are shown in Table 2.

Considering the different ways of preparing the infusion, different compositions of the prepared infusions are expected (Table 3). Interestingly, the composition of sage infusions varies significantly among producers, according to study by Zimmermann et al. [40]. Additionally, due to the preparation, i.e., direct volatilization and co-vaporization with water vapor, highly volatile compounds are lost, as emphasized by Radulescu, Chiliment, and Oprea [38].

In the case of solvent extraction, the chemical composition is affected by parameters like solvent type, time of extraction, temperature, and whether it is performed by convection or by extraction assisted with ultrasound or pressurized water [16,31,41,42,43,44,45,46,75,85,86]. Because of the multiple possibilities of changing the process parameters that have an impact on the composition of the product, the precise concentration of the components differs between studies (Table 3). Although components such as 1,8-cineole, camphor, and borneol can be found in solid–liquid extracts [86], the most commonly tested components in such extracts are polyphenolic components such as ferulic acid, rosmarinic acid, apigenin, luteolin-7-O-rutinose, carnosic acid, cinnamic acid, and quercetin-7-O-glucoside [16,41,42,43,44,45].

## 6. Sage and Health Benefits

Along with some of the traditional uses of sage mentioned in the introduction [6], many recent studies report on anti-inflammatory and antinociceptive effects related to pain relief, antioxidant and antidementia effects related to Alzheimer’s disease, antimicrobial effects related to various infections including worm infestations and gastroenteritis, anticancer and antimutagenic effects related to various cancers such as colon or breast cancer, and very important hypoglycemic and hypolipidemic effects related to metabolic diseases such as non-alcoholic fatty liver or diabetes [8,11,87].

However, the main obstacle when assessing the relevance of reported results remains the variable extracts used (e.g., tea, essential oils, ethanolic extracts, etc.), with different compositions of bioactive compounds. For example, the anti-inflammatory effect of the methanolic extract is associated with a higher content of polar components such as rosmarinic, ursolic, caffeic, and oleanolic acids [88]. In the chloroform extract, ursolic and oleanolic acid are present in the highest amounts, and they have been proven to have the best dose-dependent topical anti-inflammatory activity [89].

A lot of attention has been put on specific components found in different sage extracts that are being analyzed, mostly in vitro or in animal studies. For example, Juhás et al. [90] found that borneol, one of the key ingredients in sage essential oil, significantly suppresses pro-inflammatory cytokine mRNA expression characteristic of colonic inflammation. Carnosic acid, from the methanolic sage leaf extract, especially at a dose of 20 mg kg^−1^, significantly inhibited triglyceride elevation, reduced body weight gain, and inhibited activity against pancreatic lipase [91].

According to Ghorbania and Esmaeilizadeh [8], confirmed clinical pharmacological effects of sage on humans so far include improvement of memory and cognitive functions, pain relief, especially for sore throat, and significant improvement in blood glucose (including HbA1c and post-prandial glucose) and lipid profile (especially an increase of high-density lipoprotein, HDL).

Especially interesting are the beneficial effects on memory and cognitive functions. Every year about 8 million people are newly diagnosed with dementia, 60%–80% of all dementia is Alzheimer’s, and the highest risk for developing any dementia is among the elderly. The world’s population is getting older, so the burden of Alzheimer’s and other progressive, incurable neurodegenerative diseases is a major public health issue. Finding a way to prevent or cope with the degenerative nature of the disease is the best way we can determine striking predictions for the near future. Traditionally, sage has been used to improve memory and reduce age-related cognitive decline. Besides well-documented antioxidant effects, major components in sage have shown to decrease the inflammation resulting from the neurotoxic effects of accumulated amyloid-β peptide, which is a characteristic of Alzheimer’s disease [87]. Even the sage aroma shows a positive effect on memory [92].

Metabolic improvements in terms of glucose and lipid profile are interesting for the fact that alterations in these parameters are directly related to diabetes, obesity, non-alcoholic liver disease, metabolic syndrome, and cardiovascular diseases, i.e., diseases with the highest mortality and morbidity rates with immense individual and societal burdens. Hernandez-Saavedra et al. [93] observed the effect of sage infusion on obesity-related metabolic alterations in rats during a 12-week period. Significant reductions in the total cholesterol, triglycerides, low-density lipoprotein, and C-reactive protein was found, along with a decrease in body weight and abdominal fat mass [93]. Several other studies illustrated various positive metabolic changes in animal studies [94,95]. A non-randomized crossover trial on six women who consumed 600 mL of sage infusion per day during 4 weeks, followed by 2 weeks of a wash-out period, found no effect on plasma glucose, but the level of LDL and the total cholesterol lowered, while HDL increased [96]. Importantly, no adverse hepatotoxic effects were observed.

Colorectal cancer draws a lot of scientific interest because of its strikingly high correlation with the diet. Sage infusion has been found to prevent colorectal cancer in rats [97], but also has cytotoxic effects on cancer cell lines [11,98] and diminishes the negative effects of radiotherapy used for the cancer treatment [99].

## 7. Conclusions

Sage has been long used and has well-documented benefits for various health conditions, but continues to elicit interest by researchers around the globe. The most commonly used and tested is sage essential oil, but recent studies, especially in the field of oncology and neurodegenerative diseases, show that other sage products offer a huge potential. New extraction techniques, such as UAE and MAE or SC-CO_2_ extraction, allow us to determine new compounds in sage extracts that are unable to get by hydrodistillation or infusion. By optimization of the extraction techniques, i.e., extraction parameters, we are able to get the desired composition with the highest activity for a specific purpose. As in all other cases when the processing material is a plant, harvesting conditions, geographical area, and the plant itself affect the yield and the final bioactive composition of the extract. We are yet to evidence the number of extracts and their benefits, especially for the treatments of cancers and neurological diseases.

## Figures and Tables

**Figure 1 plants-08-00055-f001:**
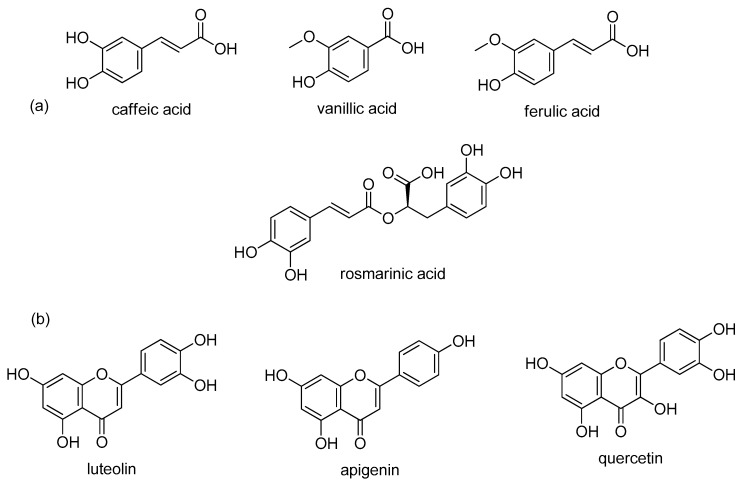
Major phenolic components in sage extract; (**a**) phenolic acids; (**b**) flavonoids.

**Figure 2 plants-08-00055-f002:**
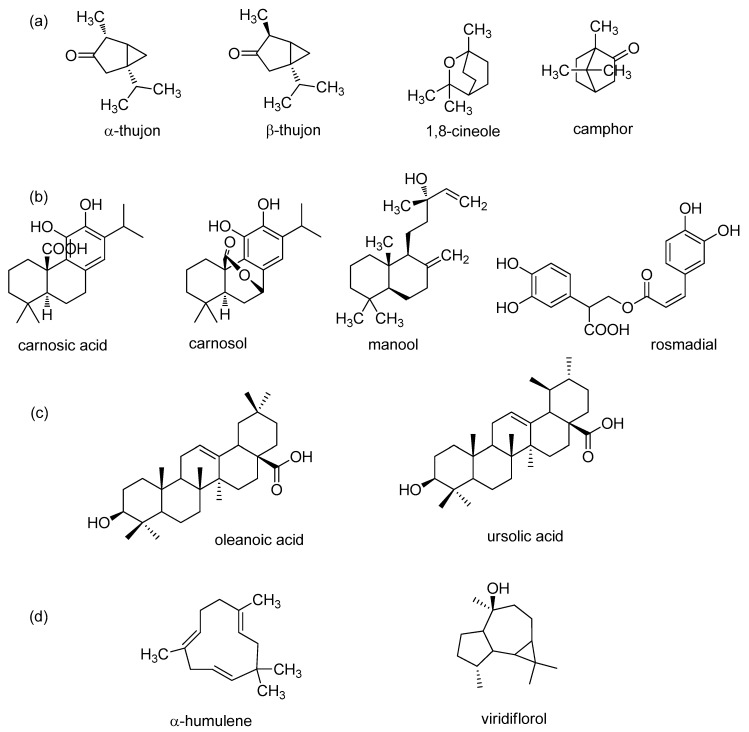
Major terpenes in sage extract; (**a**) monoterpenes; (**b**) diterpenes; (**c**) triterpenes; and (**d**) sesquiterpenes.

**Table 1 plants-08-00055-t001:** The composition of bioactive compounds found in sage extracts obtained by hydrodistillation.

Plant Part	Yield	Bioactive Compounds	Country	Reference
Herba	24.8 mL/kg	α-Pinene (3.5%), Camphene (5.3%), 1,8-Cineole (11.9%), α-Thujone (21.0%), β-Thujone (10.1%), Camphor (23.9%), Borneol (2.6%), Bornyl acetate (2.6%), (E)-b-Caryophyllene (3.4%), α-Humulene (3.3%), Viridiflorol (5.6%)	France	[17]
	10.0 mL/kg	α-Pinene (5.8%), Camphene (5.1%), 1,8-Cineole (14.6%), α-Thujone (18.6%), β-Thujone (6.6%), Camphor (13.7%), Borneol (5.0%), Bornyl acetate (1.2%), (E)-b-Caryophyllene (2.9%), α-Humulene (2.6%), Viridiflorol (8.2%)	Hungary	[17]
	15.0 mL/kg	α-Pinene (5.1%), Camphene (6.8%), 1,8-Cineole (12.6%), α-Thujone (19.6%), β-Thujone (5.4%), Camphor (19.2%), Borneol (2.0%), Bornyl acetate (1.7%), (E)-b-Caryophyllene (1.1%), α-Humulene (1.4%), Viridiflorol (10.4%)	Belgium	[17]
	12.8 mL/kg	α-Pinene (4.4%), Camphene (7.1%), 1,8-Cineole (17.0%), α-Thujone (16.2%), β-Thujone (7.1%), Camphor (28.5%), Borneol (2.8%), Bornyl acetate (1.9%), (E)-b-Caryophyllene (0.9%), α-Humulene (2.0%), Viridiflorol (4.5%)	Russia	[17]
	21.8 mL/kg	α-Pinene (5.1%), Camphene (5.9%), 1,8-Cineole (45.3%), α-Thujone (3.0%), β-Thujone (1.5%), Camphor (11.3%), Borneol (1.6%), Bornyl acetate (0.1%), (E)-b-Caryophyllene (4.9%), α-Humulene (0.4%), Viridiflorol (1.1%)	Greece	[17]
	21.1 mL/kg	α-Pinene (3.7%), Camphene (3.6%), 1,8-Cineole (1.6%), α-Thujone (13.7%), β-Thujone (11.6%), Camphor (12.9%), Borneol (3.0%), Bornyl acetate (1.9%), (E)-b-Caryophyllene (2.7%), α-Humulene (2.1%), Viridiflorol (7.9%)	Ukraine	[17]
	4.2 mL/kg	α-Pinene (0.2%), Camphene (0.2%), 1,8-Cineole (9.1%), α-Thujone (6.8%), β-Thujone (1.6%), Camphor (29.8%), Borneol (11.8%), Bornyl acetate (7.8%), (E)-b-Caryophyllene (1.6%), α-Humulene (1.8%), Viridiflorol (4.5%)	Scotland	[17]
	2.2 mL/kg	α-Pinene (0.1%), Camphene (0.1%), 1,8-Cineole (2.7%), α-Thujone (18.7%), β-Thujone (11.7%), Camphor (12.7%), Borneol (2.4%), Bornyl acetate (1.9%), (E)-b-Caryophyllene (7.5%), α-Humulene (7.5%), Viridiflorol (15.7%)	Moldavia	[17]
	5.1–15.2 mL/kg	α-Pinene (0.6–6.4%), Camphene (0.6–5.5%), 1,8-Cineole (5.3–14.6%), α-Thujone (15.2–26.6%), β-Thujone (5.2–12.9%), Camphor (16.4–20.0%), Borneol (1.8–4.9%), Bornyl acetate (2.1–2.2%), (E)-b-Caryophyllene (2.4–4.5%), α-Humulene (5.3–8.5%), Viridiflorol (4.0–8.5%)	Estonia	[17]
Leaves, dried	1.4%–3.5%	α-Pinene (5.73–6.64%), Camphene (6.16–8.13%), β-Pinene (1.42–2.68%), Myrcene (1.01–1.27%), Limonene (1.82–2.63%) 1,8-Cineole (8.95–10.43%), α-Thujone (23.61–26.17%), β-Thujone (3.91–4.38%), Camphor (20.50–23.14%), Linalool (0.37–2.40%), Bornyl acetate (2.12–2.3.52%), Isoborneol (0.04–2.80%), Borneol (9.99–11.03%)	Croatia (mainland)	[20]
	1.4%–3.5%	α-Thujene (0.18–1.38%), α-Pinene (2.65–4.90%), Camphene (2.40–8.48%), β-Pinene (1.07–3.38%), Myrcene (0.46–1.57%), Limonene (1.03–3.64%), 1,8-Cineole (7.84–22.46%), α-Thujone (7.17–36.33%), β-Thujone (3.94–31.89%), Camphor (6.99–19.61%), Linalool (0.32–4.66%), Bornyl acetate (0.59–5.32%), Isoborneol (0.14–2.12%), Borneol (6.45–15.54%)	Croatia (island)	[20]
	2.4%–3.2%	α-Pinene (5.4–6.6%), Camphene (4.2–5.3%), β-Pinene (2.6–3.3%), Limonene (1.3–1.7%), 1,8-Cineole (13.4–16.8%), α-Thujone (1.1–1.5%), β-Thujone (15.0–17.7%), Camphor (27.0–32.2%), Bornyl acetate (0.9–1.8%), β-Caryophyllene (3.5–4.3%), Terpinen-4-ol (0.1–1.1%), α-Humulene (0.9–1.8%), Borneol (1.7–3.7%), iso-Borneol (0.3–2.0%), Ledol (1.7–2.8%)	Portugal	[27]
	2.2%–3.1%	α-Pinene (1.9–2.7%), Camphene (3.9–5.9%), β-Pinene (1.1–1.6%), Limonene (0.6–1.5%), 1,8-Cineole (7.1–9.7%), α-Thujone (19.6–24.3%), β-Thujone (1.7–2.9%), Camphor (23.8–27.9%), Bornyl acetate (2.8–3.9%), β-Caryophyllene (1.7–2.4%), α-Humulene (7.6–12.4%), Borneol (2.9–4.3%), iso-Borneol (0.3–2.0%), Ledol (3.7–7.5%), Caryophyllen-8-ol (1.1–2.6%)	Hungary	[27]
	2.0%–2.5%	α-Pinene (5.7–9.0%), Camphene (2.2–2.9%), β-Pinene (2.3–4.4%), Limonene (0.8–1.4%), 1,8-Cineole (3.6–6.0%), (E)-β-Ocimene (0.1–1.3%), α-Thujone (18.9–26.6%), β-Thujone (5.0–8.3%), Camphor (18.2–27.3%), Linalol (0.3–1.4%), Bornyl acetate (0.6–1.4%), β-Caryophyllene (1.0–1.6%), Terpinen-4-ol (0.4–1.2%), α-Humulene (7.3–10.5%), Borneol (1.8–3.3%), Ledol (1.8–3.6%),Caryophyllen-8-ol (1.3–4.3%)	Romania	[27]
	1.8%–2.7%	α-Pinene (1.0–1.3%), Camphene (2.1–3.8%), β-Pinene(1.1–1.6%), Limonene (1.0–1.4%), 1,8-Cineole (9.5–13.3%), β-Thujone (24.4–25.9%), Camphor (20.8–27.1%), β-Caryophyllene (4.8–8.0%), α-Humulene (3.5–5.2%), Thujyl alcohol (1.1–1.3%), Borneol (2.8–3.7%), Ledol (4.9–6.8%), Caryophyllen-8-ol (0.9–1.6%)	Czech Republic	[27]
	1.3%–2.5%	α-Pinene (0.8–2.9%), Camphene (2.0–3.2%), β-Pinene (1.0–2.8%), Limonene (0.5–1.5%), 1,8-Cineole (8.6–11.6%), α-Thujone (22.2–31.9%), β-Thujone (2.7–8.9%), Camphor (15.8–24.0%), Bornyl acetate (0.4–2.8%), β-Caryophyllene (1.0–4.4%), α-Humulene (5.5–7.6%), Borneol (1.9–4.5%), Ledol (3.0–4.1%), Caryophyllen-8-ol (1.9–4.1%)	France	[27]
	/	α-Pinene (4.9%), Camphene (5.0%), β-Pinene (3.4%), 1,8-Cineole (12.1%), α-Thujone+linalool (21.2%), β-Thujone (4.4%), Camphor (23.6%), Borneol (5.6%), E-caryophyllene (2.7%), α-Humulene (5.2%), Viridiflorol (3.0%)	Estonia	[21]
	/	α-trans-Ocimene (1.69%), Camphene (1.66%), 1-Octen-3-ol (8.50%), 1,8-Cineole (6.72%), α-Thujone (21.85%), β-Thujone (5.51%), Camphor (11.25%), 1-Borneol (2.58%), Bornyl acetate (3.22%), β-Caryophyllene (3.54%), α-Humulene (4.51%), α-Farnesene (1.15%), Viridiflorol (11.71%), Citronellyl propionate (1.22%), Manool (9.15%)	Romania	[38]
	1.02%	α-Thujone (19.02%), Viridiflorol (18.96%), 1,8-Cineole (8.58%), Limonene (6.56%), trans-Carryophyllene (5.20%), β-Thujone (4.09%), α-Thujene (3.42%), β-Pinene (2.19%), Camphor (2.10%), Linalool (2.02%)	Tunisia	[58]
	/	α-Pinene (4.5%), Camphene (2.8%), β-Pinene (1.5%), 1,8-Cineole (14.1%), Thujone (56.5%), Camphor (5.7%), α-Humulene (6.9%)	Croatia	[22]
	27 mL/kg	α-Pinene (4.35%), Camphene (7.61), *p*-Cymene (2.77%), 1,8-Cineole (7.96%), α-Thujone (24.29%), β-Thujone (4.03%), Camphor (23.72%), Borneol (2.21%), Bornyl acetate (2.73%), β-Caryophyllene (2.25%), α-Humulene (2.83%), Viridiflorol (6.41%), Manool (4.07%)	Albania	[34]
	1.08%–1.37%	α-Pinene (8.26%), Camphene (7.27%), 1,8-Cineole (20.13%), cis-Thujone (26.85%), Camphor (16.66%), Borneol (2.76%), Bornyl acetate (1.90%), E-Caryophyllene (1.31%), α-Humulene (1.83%), Viridiflorol (1.08%)	Croatia	[30]
	97 (±3.7) mg/kg leaf dry weight	α-Pinene (1.1 ± 0.09%), Camphene (2.3 ± 0.18%), β-Pinene (1.6 ± 0.32%), Myrcene (1.4 ± 0.11%), Limonene (1.3 ± 0.03%), 1,8-cineole (10.4 ± 1.79%), α-Thujone (17.3 ± 2.94%), Camphor (29.2 ± 2.84%), β-Thujone (4.9 ± 0.64%), β-Caryophyllene (6.4 ± 1.21%), α-Humulene (3.7 ± 1.94%), Caryophyllene oxide (1.9 ± 0.78%), Viridiflorol (11.6 ± 2.23%)	Poland	[33]
Leaves, fresh, completely and incompletely developed	0.3%–2.9%	Camphor (7.0–32.7%), cis-Thujone (6.7–20.0%), α-Humulene (3.4-18.9%), Viridiflorol (5.7–12.4%), Manool (1.4–14.5%), Camphene (3.6–8.6%), 1,8-Cineole (3.0–6.9%), Limonene (2.2–9.1%), β-Pinene (2.7–13.5%), trans-Thujone (0.7–2.4%), α-Pinene (3.4–5.2%), Myrcene (0.6–1.2%), cis-β-Ocimene (0.0–3.2%), Borneol (1.3–3.0%), Bornyl acetate (0.1–1.7%), β-Caryophyllene (1.0–4.7%)	Serbia	[18]
	0.2%–2.1%	Camphor (1.9–30.4%), cis-Thujone (10.6–28.5%), α-Humulene (4.5–33.3%), Viridiflorol (2.9–10.7%), Manool (1.7–9.2%), Camphene (0.2–3.6%), 1,8-Cineole (1.2–19.4%), Limonene (0.5–3.6%), β-Pinene (0.4–6.5%), trans-Thujone (1.4–14.5%), α-Pinene (0.4–1.9%), Myrcene (0.6–1.3%), cis-β-Ocimene (0.0–4.8%), trans- β-Ocimene (0.0–1.7%), Borneol (0.3–1.9%), β-Caryophyllene (0.4–2.5%), α-Terpinene (0.1–1.5%), *p*-Cymene (0.3–3.6%), γ-Terpinene (0.4–2.9%), cis-Sabinene hydrate (0.4–1.5%), cis-Pinocamphone (0.0–1.9%), Terpinen-4-ol (0.1–2.2%)	Croatia	[18]
Leaves and flowers	1.59%–1.87%	α-Pinene (3.54%), Camphene (5.63%), Myrcene (5.47%), 1,8-Cineole (19.6%), Camphor (46.1%), Borneol (4.54%), Viridiflorol (0.26%)	Spain	[28]
Dried aerial parts	2.0%–2.1%	α-Pinene (6.5–8.2%), Camphene (2.4–2.9%), β-Pinene (2.8–3.4%), Myrcene (2.0–2.1%), *p*-Cymene (1.5–1.7%), 1,8-Cineole (64.3–67.1%), α-Thujone (1.2–1.4%), β-Thujone (2.3–2.8%), Camphor (5.3–6.1%), α-Terpineol (1.0–1.2%), β-Caryophyllene (1.4–1.6%)	Portugal	[14]
	4.0%	Cineole (13.69%), Borneol (13.77%), α-Thujone (12.46%), Ledene (11.05%), β-Pinene (7.00%), α-Humulene (6.92%), Trans-caryophyllene (5.28%), β-Thujone (4.56%), Camphor (3.58%), Naphthalene (3.27%), Camphene (2.86%), Bicyclo (1.75%)	Iran	[100]
	2.13%–3.3%	α-Thujone (0.300–0.378 µgg^−1^), Camphor (5.88–16.3 µgg^−1^), β-Thujone (0.300–0.378 µgg^−1^), Carvacrol (2.28–51.1 µgg^−1^), 1,8-Cineole (20.1–37.9 µgg^−1^)	Jordan	[24]
	0.58%	1,8-Cineole (33.27%), β-Thujone (18.40%), α-Thujone (13.45%), Borneol (7.39%), β-Elemene (4.82%), Camphor (3.31%), α-Pinene (2.74%), Fenchyl acetate (1.6%), α-Muurolol (1.41%), Camphene (1.03%),	Tunis	[101]
	1.1%–1.2%	α-Pinene (1.05–1.63%), β-Pinene (1.81–3.80%), Myrcene (1.00–1.07%), 1,8-Cineole (8.85–15.6%), α-Thujone (11.55–19.23%), β-Thujone (5.45–6.17%), Camphor (5.08–15.06%), Borneol (1.35–2.87%), β-Caryophyllene (2.63–9.24%), α-Humulene (1.93–8.94%), Viridiflorol (9.94–19.46%), Manool (5.52–13.06%)	Tunisia	[36]
	0.5%	α-Pinene (4.60%), Camphene (3.61%), β-Pinene (1.18%), Limonene (1.52%), 1.8-Cineole (10.03%), β-Thujone (33.15%), α-Thujone (8.73%), Camphor (13.16%), Borneol (2.98%), Bornyl acetate (1.02%), Viridiflorol (3.13%), Humuleneepoxide II (1.21%), Manool (1.48%)	Croatia	[29]
Fresh aerial part	0.29%–0.39%	α-Pinene (0.27–3.86%), Camphene (1.14–5.65%), β-Pinene (0.23–2.02%), Limonene (tr.-1.42%), Eucalyptol (4.98–13.4%), α-Thujone (35.9–45.8%), β-Thujone (4.35–9.60%), Camphor (15.5–21.1%), Borneol (0.74–3.20%), β-Caryophyllene (tr.-3.78%), α-Humulene (tr.-3.85%)	Italy	[78]
Fresh plant	/	Cis-Salvene (1.46%), α–Pinene (4.12%), Camphene (2.87%), β-Pinene (6.06%), Eucalyptol (11.17%), Thujone (35.86%), Camphor (8.13%), Terpinen-4-ol (1.53%), α-Terpineol (1.09%), Linalylacetate (1.87%), α-Humulene (3.25%), Cadinene (1.04%)	Greece	[35]

**Table 2 plants-08-00055-t002:** The composition of bioactive compounds found in sage extracts obtained by supercritical fluid extraction.

Plant Part	Extraction Parameters	Yield	Bioactive Compounds	Country	Reference
Leaves, dried	pressure 15 MPa, temperature 40 °C, CO_2_ flow rate 0.48-0.53 kg h^−1^, time 1.70–1.82 h	4.453%	α-Pinene (1.80%), Camphene (1.54%), 1,8-Cineole (6.57%), cis-Thujone (10.03%), Camphor (10.76%), α-Humulene (3.90%), Viridiflorol (7.70%), Manool (17.70%), Labda-7,14-diene-13-ol (2.97%), Abietol (2.36%), Heneicosane (1.02%), Octacosane (2.77%), Triacontane (4.43%)	Croatia	[30]
	pressure 25 MPa, temperature 60 °C, CO_2_ flow rate 6 kg h^−1^, time 90 min, co-solvent (95% ethanol - 2% *w*/*w*)	90 (±7.5) mg kg^−1^ leaf dry weight	α-Pinene (1.0 ± 0.10%), Camphene (1.4 ± 0.58%), 1,8-Cineole (4.6 ± 0.95%), α-Thujone (7.5 ± 1.07%), β-Thujone (4.9 ± 0.88%), Borneol (8.4 ± 1.53%), Menthol (1.3 ± 0.37%), Camphor (16.4 ± 1.11%), Bornyl acetate (2.2 ± 0.58%), α-Humulene (6.4 ± 1.36%), Viridiflorol (22.51 ± 1.99%), Humuleneepoxide II (2.4 ± 0.49%), β-Caryophyllene (6.4 ± 1.53%), Caryophylleneoxide (1.5 ± 0.80%)	Poland	[33]
	pressure 30 MPa, temperature 40 °C CO_2_ flow rate 2.4 kg h^−1^, time 1.5–4.5 h, fractionation in 2 separators (s1, s2)	1.39% 3.23%	s1 = 1,8Cineole (13.97–14.32%), Cis sabinene hydrate (-,1.17%), Linalool (-,1.79%), Cissabinol (-,2.24%), α-Terpineol (-,1.39%), Geraniol (-,1.48%), Camphor (43.46–59.03%), Borneol (6.91–14.08%), Linalyl acetate (-,6.48%), Endobornyl acetate (-,4.70%) Sabinyl acetate (5.15–12.92%), α-Terpinenyl (-,3.36%), E-Caryophyllene (-,2.31%), Humulene (-,1.56%), Geranylpropionate (-,1.91%), Spathulenol (-,1.63%), Viridiflorol (2.29%); s2 = 1,8-Cineole (4.27–17.12%), Trans Sabinenehydrate (0.50–10.92%), Linalool (-,1.34%), Cissabinol (2.37–3.16%), Camphor (30.79–43.07%), Borneol (7.29–12.50%), α-Terpineol (1-40-3.10%), Geraniol (1.16,3.0%), Linalyl acetate (2.65–4.78%), Endobornyl acetate (1.65–3.21%), Sabinyl acetate (4.84–23.90%), α-Terpinenyl (-,3.46%), E-Caryophyllene (1.98–2.56%), Humulene (-,1.42%), Geranylpropionate (-,1.30%), Spathulenol (-,2.45%), Viridiflorol (1.98–5.42%)	Spain	[57]
	pressure (80, 100, 150, 200, 300 MPa), temperature 40 °C, CO_2_ flow rate 3.23 × 10^−3^ kg min^−1^, time 4 h	0.76%–4.65%	α-Thujone (0.66–5.15%), Camphor (1.43–15.24%), Isoborneole (6.80–11.29%), Terpineol-L-4 (0.25–2.08%), Bornyl–acetate (2.01–5.90%), Sabinyl-acetate (0.41–1.05%), Isocaryophyllene (0.84–2.74%), α-Gurjunene (0.44–1.45%), γ-Elemene (7.02–24.98%), Selina-3,7(11) diene (11.25–13.83%), 1,11-Epoxyhumulene (1.98–3.67%), Caryophylleneoxide (0.87–1.73%), Phyllocladene (4.19–23.37%)	Bosnia and Herzegovina	[59]
	pressure 80–100 MPa, temperature 45–60 °C, CO_2_ flow rate 0.95 kg h^−1^, 2 separators	1.35%	α-Pinene (2.37%), Camphene (1.02%), β-Pinene (2.44%), β-Myrcene (2.29%), Cymene-orrho (1.82%), 1,8-Cineole (54.36%), α-Thujone (1.38%), β-Thujone (1.42%), Camphor (5.74%), α-Terpineol (1.61%), Caryophyllene (7.06%), α-Humulene (1.27%), β-Bisabolene (1.04%), y-Cadinene (1.46%), Manool (1.79%), 1,8-Cineole (54.36%), Camphor (5.74%), Caryophyllene (7.06%), α-Pinene (2.37%) β-Pinene (2.44%), β-Myrcene (2.29%)	Italy	[55]
	pressure 17.2 MPa, temperature 45 °C, CO_2_ flow rate 1 mL min^−1^, time 60 min	13.2%, 7.6%	α-Pinene (5.3%), Camphene (6.1%), β-Pinene (9.5%), 1,8-Cineole (9.7%), α-Thujone+linalool (27.1%), β-Thujone (4.4%), Camphor (15.6%), Borneol (2.1%), E-caryophyllene (2.2%), α-Humulene (4.9%), Viridiflorol (1.6%)	Estonia	[21]
	pressure 9 MPa, temperature 25 and 50 °C, CO_2_ flow rate 0.35 g min^−1^, time 3 h	2.7%–4.8%	α-Pinene (1.45, 4.28%), Camphene (2.54, 7.16%), β-Myrcene (1.06, 1.11%), *p*-Cymene (1.22, 3.14%), 1,8-Cineole (3.45, 9.54%), α-Thujone (17.49, 26.52%), β-Thujone (2.54, 4.23%), Camphor (19.08, 27.26%), Borneol (2.34%, tr.), Bornyl acetate (2.00, 2.25%), β-Caryophyllene (4.06, 3.83%), β-Gurjunene (1.00%, tr.), Aromadendrene (1.34, tr.), α-Humulene (4.73, 3.82%), Viridiflorol (6.64%, -), Manool (15.28, 0.65%), Sclareol (0.71, 1.52%), Heneicosane (3.92, 2.04%), Hentriacontane (4.66%, tr.)	Albania	[34]
	pressure 65–160 MPa, temperature 50 °C, flow rate 3.5–4 g min^−1^, time 5 h	/	Manool (32.39–56.49%), Ledene (4.43–7.63%), Viridiflorol (4.50–24.69%), 5,8-Dimethoxy-2-methyl-4*H*-Naphtho[2,3-b]pyran-4,6,9-trione (0.03–7.18%) Camphor (1.00–4.45%), Estra-1,3,5(10),9(11)-teraen-17-one (0.02–2.84%), β-Caryophyllene (1.13–2.33%), 1,8-Cineole (0.80–1.79%), α-Thujone (1.00–1.57%), Aromadendrene (0.65–1.01%)	Tunisia	[58]
	pressure 10–30 MPa, temperature 40–60 °C, CO_2_ flow rate 1–3 kg h^−1^, time 90 min	0.242%–7.361%	1,8-Cineole (6.56–25.52 mg CE g^−1^), α-/β-Thujone (11.56–34.68 mg CE g^−1^), Camphor (38.23–102.97 mg g^−^^1^), α-Humulene (39.90–90.73 mg CE g^−^^1^), Viridiflorol (48.07–97.01 mg CE g^−1^), Manool (113.90–335.36 mg CE g^−1^), α-Pinene (0.47–9.09 mg CE g^−1^), Camphene (0.35–7.87 mg CE g^−1^), β-Pinene (0.24–1.15 mg CE g^−1^), β-Myrcene (0.24–1.15 mg CE g^−1^), *p*-Cymene (0.24–1.15 mg CE g^−^^1^), Limonene (0.24–1.68 mg CE g^−^^1^), Linalool (0.24–2.30 mg CE g^−^^1^), Borneol (7.32–31.01 mg CE g^−1^), Terpinen-4-ol (0.24–2.30 mg CE g^−1^), p-Cymen-8-ol (0.24–1.35 mg CE g^−1^), α-Terpineol (0.48–1.35 mg CE g^−1^), Myrtenol (0.00–1.15 mg CE g^−1^), Bornyl acetate (3.54–14.93 mg CE g^−1^), trans-β-caryophyllene (3.54–16.08 mg CE g^−1^), 6-Oxobornyl acetate (3.07–14.93 mg CE g^−1^), Alloaromadendrene (0.24–2.30 mg CE g^−1^), Ledene (0.47–2.30 mg CE g^−1^)	Croatia	[60]
Leaves and flowers	pressure 90 and 100 MPa, temperature 40 and 50 °C, particle diameter (0.3, 0.5, 0.8 mm), CO_2_ flow rate (0.72, 1.02, 1.32 kg h^−1^	1.27%–1.88%	α-Pinene (1.33,1.54%), Camphene (1.73,2.42%), Myrcene (2.65,3.89%), 1,8-Cineole (16.1,14.2%), Camphor (40.9,48.0%), Borneol (4.62,4.17%), α-Terpineol (1.45, 0.95%), β-Caryophyllene (1.47,1.53%), Methyldodecanoate (1.77,1.85%), Viridiflorol (1.41,0.24%)	Spain	[28]
Dried aerial parts	pressure 7,10,15,20,30 MPa, temperature 50 °C, CO_2_ flow rate 0.4 kg h^−1^	4.82%	α-Pinene (0.77–2.07%), Camphene (0.50–1.54%), 1.8-Cineole (1.88–4.75%), *β*-Thujone (7.95–16.56%), α-Thujone (3.26–8.12%), Camphor (7.95–10.64%), Neo-3-thujanol (2.16–2.51%), Myrtenol (0.66–1.06%), α-Campholenicacid (1.43–2.52%), Acetophloroglucine (0.72–2.54%), Trans-caryophyllene (1.39–2.16%), α-Humulene (1.90–3.08%), Caryophylleneoxide (0.60–2.62%), Viridiflorol (4.14–9.58%), Humuleneepoxide II (1.16–4.78%), Muurola-4.10(14)-dien-1-β-ol (0.50–1.04%), Manool (13.15–21.75%), Carnosol derivative (6.25–13.09%), Trans-ferruginol (0.08–1.71%), Methylhexadecanoate (0.74–1.98%), Methyloleate (0.07–1.19%), Methyloctadecanoate (0.07–2.46%), Octacosane (0.35–2.17%), Untriacontane (0.00–1.33%), Olean-18-ene (0.52–4.24%), Lupeol (0.70–3.04%)	Croatia	[29]

**Table 3 plants-08-00055-t003:** The composition of bioactive compounds found in sage extracts, obtained by classic extraction techniques.

Type of Extraction	Plant Part	Extraction Parameters	Yield	Bioactive Compounds	Country	Reference
INFUSION	Flowering aerial parts		/	Luteolin diglucuronide (11.89 ± 0.15 mg g^−1^), 6-Hydroxyluteolin 7-O-glucuronide (2.53 ± 0.08 mg g^−1^), Sagecoumarin (1.11 ± 0.05 mg g^−1^), Luteolin 7-O-rutinoside (9.35 ± 0.20 mg g^−1^), Luteolin 7-O-glucuronide (88.12 ± 0.36 mg g^−1^), Luteolin 7-O-glucoside (37.41 ± 0.65 mg g^−1^), Sagerinic acid (2.92 ± 0.08 mg g^−1^), cis-Rosmarinic acid (0.97 ± 0.07 mg g^−1^), trans-Rosmarinic acid (73.97 ± 0.15 mg g^-1^), Apigenin 7-O-glucoside (5.40 ± 0.01 mg g^−1^), Luteolin acetylglucoside (15.56 ± 0.33 mg g^−1^), Hispidulin glucuronide (10.53 ± 0.25 mg g^−1^), Hispidulin (1.01 ± 0.03 mg g^−1^)	Spain	[39]
	Teas (commercial brands) or sage leaves		/	Saponin (3.8 ± 0.77-12.9 ± 0.25 mg L^−1^), Luteolin-diglucuronide (5.1 ± 1.20-44.0 ± 1.99 mg L^−1^), Hydroxyluteolin-glucuronide (6.7 ± 1.57 mg L^−1^), Apigenin-diglucuronide (1.1 ± 0.22-9.1 ± 0.05 mg L^−1^), Luteolin-7-O-glucoside (3.5 ± 0.71-8.4 ± 0.22 mg L^−1^), Luteolin-rutinoside (4.9 ± 0.31-10.7 ± 0.60 mg L^−1^), Luteolin-7-O-glucuronide (37.9 ± 2.17-166.3 ± 1.65 mg L^−1^), Rosmarinic acid (30.5 ± 1.00-295.7 ± 9.71 mg L^−1^), Apigenin-glucuronide (8.6 ± 0.39-41.1 ± 1.15 mg L^−1^), Salvianolic acid K (6.8 ± 0.18-56.4 ± 2.95 mg L^−1^), Carnosic acid (9.1 ± 1.20-32.9 ± 3.71 mg L^−1^)	Germany	[40]
MACERATION	Leaves, dried	methanol, 72 h, room temperature	23.41% ± 2.65%	Chlorogenic acid (1.22%), Caffeic acid (1.98%), Quinic acid (1.19%), *p*-Coumaric acid (1.2%), Caffeoyl quinic acid derivative (1.07%), Quercetin-7-O-glucoside (2.52%), Ferulic acid (18.79%), Carnosic acid (3.77%), Cinnamic acid (2.57%), Rosmarinic acid (17.85%), Apigenin (14.32%), Luteolin-7-O-rutinose (8.61%)	Egypt	[44]
SOXHLET	Leaves, dried (commercial samples)	hexane and ethyl acetate, 6 h	/	Rosmarinic acid (10.0 ± 0.92 mg g^−1^), Apigenin (2.5 ± 0.38 mg g^−1^), Hispidulin (6.3 ± 0.58 mg g^−1^), Carnosol (31.1 ± 1.00 mg g^−1^), Rosmadial (6.8 ± 0.42 mg g^−1^), Carnosic acid (42.9 ± 3.05 mg g^−1^), Methyl carnosate (8.6 ± 0.22 mg g^−11^), Oleanolic acid (171.9 ± 10.6 mg g^−1^), Ursolic acid (358.8 ± 14.2 mg g^−1^)		[37]
	Dried aerial parts	ethanol and water (70:30 *v*/*v*), 4 h	26.5%	α-Campholenic acid (1.21%), Cis-α-Bergamotene (1.61%), Viridiflorol (4.25%), Humuleneepoxide II (1.31%), Manool (13.15%), Carnosol derivative (15.21%), Trans-ferruginol (1.49%), Methylhexadecanoate (1.08%), Heptacosane (1.10%), Nonacosane (6.21%), Untriacontane (7.05%), t-Sitosterol (1.25%), Olean-18-ene (24.76%), Lupeol (8.01%)	Croatia	[29]
		methanol, 2 h under nitrogen atmosphere	/	Caffeic acid (222.24 ± 11.23-695.04 ± 18.21 µg g^−1^ DM), Ferulic acid (312.43 ± 2.53-703.29 ± 17.74 µg g^−1^ DM), Rosmaric acid (13,680.22 ± 101.77-18,378.00 ± 393.26 µg g^−1^ DM), Gallic acid (14.49 ± 2.41-29.74 ± 1.05 µg g^−1^ DM), p-hydroxy benzoic acid (121.15 ± 2.16-122.31 ± 2.65 µg g^−1^ DM), Carnosic acid (3278.30 ± 227.59-6001.75 ± 390.12 µg g^−1^ DM), Carnosol (5045.42 ± 318.10-5947.03 ± 173.45 µg g^−1^ DM), Methyl carnosate (4816.59 ± 199.40-7174.00 ± 73.27 µg g^−1^ DM), Luteolin-7-O-glucoside (386.63 ± 0.39-661.04 ± 65.60 µg g^−1^ DM), Apigenin-7-glucoside (210.01 ± 0.70-913.90 ± 166.89 µg g^−1^ DM), Luteolin (21.41 ± 0.54-66.44 ± 1.90 µg g^−1^ DM), Apigenin (55.77 ± 6.65-77.51 ± 4.22 µg g^−1^ DM), Genkwanin (21.57 ± 0.80-25.60 ± 4.58 µg g^−1^ DM), Naringin (485.77 ± 32.41-857.92 ± 8.41 µg g^−1^ DM)	Tunisia	[36]
EXTRACTION	Herba, dried	ethanol, 1 h–7 days, Temperature 20, 30, 50 °C	/	Cineole (6.8–43.3 mg kg^−1^), Thujone (48.2–269.2 mg kg^−1^), Borneol (2.5–7.6 mg kg^−1^)	Slovakia	[45]
	Herba, dried	ethanol, stirring with and without ultrasound, temperature 20 °C	/	Cineole (14.4–33.4 mg kg^−1^), Thujone (95.0–232.9 mg kg^−1^), Borneol (5.3–8.8 mg kg^−1^)	Slovakia	[45]
	Herba, dried	ethanol ultrasound, 1 h–7 days, temperature 20, 30, 50 °C	/	Cineole (9.8–40.3 mg kg^−1^), Thujone (63.9–258.2 mg kg^−1^), Borneol (2.9–7.2 mg kg^−1^)	Slovakia	[45]
	Commercially available plant samples	methanol, acidification and elution through column,	/	Syringic acid, *p*-Coumaric acid, Ferulic acid, Sinapic acid, Luteolin, Apigenin		[85]
	Flowering aerial parts	methanol:water (80:20, *v*/*v*), 1 h, temperature 25 °C	/	Caffeic acid (2.00 ± 0.01 mg g^−1^), Luteolin diglucuronide (4.94 ± 0.01 mg g^−1^), 6-Hydroxyluteolin 7-O-glucuronide (1.72 ± 0.09 mg g^−1^), Sagecoumarin (0.76 ± 0.09 mg g^−1^), Luteolin 7-O-rutinoside (12.57 ± 0.03 mg g^−1^), Luteolin 7-O-glucuronide (94.73 ± 2.55 mg g^−1^), Luteolin 7-O-glucoside (56.09 ± 3.45 mg g^−1^), Sagerinic acid (3.35 ± 0.31 mg g^−1^), cis-Rosmarinic acid (1.20 ± 0.01 mg g^−1^), trans-Rosmarinic acid (93.22 ± 0.12 mg g^−1^), Apigenin-7-O-glucoside (7.47 ± 0.06 mg g^−1^), Luteolin acetylglucoside (21.73 ± 0.78 mg g^−1^), Hispidulin glucuronide (15.08 ± 0.14 mg g^−1^), Apigenin acetylglucoside (7.47 ± 0.06 mg g^−1^), Hispidulin (2.24 ± 0.13 mg g^−1^)	Spain	[39]
	Leaves, dried	30, 50 or 70% aqueous ethanol, acetone and water, 30, 60, 90 min, Temperature 60, 90 °C	/	Vanillic (3.04 ± 0.11-14.01 ± 0.80 mg 100 g^−1^ DM), Caffeic (8.05 ± 0.22-125.31 ± 8.32 mg 100 g^−1^ DM), Syringic (41.20 ± 0.89-70.32 ± 1.78 mg 100 g^−1^ DM), Rosmarinic (1759.10 ± 12.02-3634.12 ± 33.30 mg 100 g^−1^ DM), Salvianolic K (18.10 ± 1.00-50.14 ± 0.97 mg 100 g^−1^ DM), Salvianolic I acids (12.45 ± 0,82-26.12 ± 0.97 mg 100 g^−1^ DM), Methyl rosmarinate (9.06 ± 1.60-100.01 ± 5.47 mg 100 g^−1^ DM), 6-Hydroxyluteolin-7-glucoside (38.10 ± 2.00- 202.13 ± 0.89 mg 100 g^−1^ DM), Luteolin-7-glucuronide (109.63 ± 10.99- 356.20 ± 25.60 mg 100 g^−1^ DM), Luteolin-7-glucoside (23.65 ± 1.45-233.23 ± 10.73 mg 100 g^−1^ DM), Luteolin-3-glucuronide (470.31 ± 5.43-998.12 ± 20.01 mg 100 g^−1^ DM), Apigenin-7-glucuronide (59.55 ± 5.22-291.10 ± 0.65 mg 100 g^−1^ DM), Apigenin-7-glucoside (32.12 ± 1.65-147.26 ± 1.30 mg/100 g^−1^ DM)	Croatia	[41]
	Leaves, infusion	Infusion and elution through column, dichloromethane	/	1,8-Cineole (16.16%), α-Thujone (25.78%), β-Thujone (7.07%), Camphor (24.94%), 1-Borneol (5.38%), exo-2-Hydroxycineole acetate (1.73%), 6-Oxobornyl acetate (7.99%)	Romania	[38]
	Leaves	Liquid-liquid extraction of infusion, dichloromethane; hexane	/	1,8-Cineole (10.66; 11.37%), α-Thujone (22.61;23.95%), β-Thujone (4.58; 5.13%), Camphor (25.88; 24.93%), 1-Borneol (5.66; 5.62%), Dihydrocamphene carbinol (1.74; 0.81%), Bornyl acetate (1.02; 0.59%), 6-Oxobornyl acetate (9.18; 10.59%), Shyobunol (3.29; 0.73%)	Romania	[38]
	Leaves	Microwave power (500, 600, 700 W), 30, 50 or 70% aqueous ethanol, acetone (*v*/*v*) and water, 3, 5, 7, 9 11 min temperature 80 °C	/	Rosmarinic acid (7.1 ± 0.2-32.3 ± 0.6 mg g^−1^), sum of vanillic, caffeic, syringic, sagerinic, salvianolic acid K and salvianolic acid I (1.8 ± 0.1-2.6 ± 0.1 mg g^−1^), 6-hidroxyluteolin-7-glucoside (1.8 ± 0.1-2.5 ± 0.1 mg g^−1^), luteolin-3′-glucuronide (3.8 ± 0.1-7.5 ± 0.2 mg g^−1^), sum of luteolin-7-glucuronide, luteolin-7-glucoside, apigenin-7-glucuronide and apigenin-7-glucoside (1.6 ± 0.1-6.1 ± 0.1 mg g^−1^)	Croatia	[47]
	Leaves	Microwave power (100–500 W), 30, 50 or 70% aqueous ethanol, acetone (*v*/*v*) and water, 3, 5, 7, 9, 10 min, temperature 30, 50, 60, 80 °C, addition of 10% HCl	/	Rosmarinic acid (384.35 ± 30.78-1521.54 ± 38.44 mg 100 g^−1^), Methyl rosmarinate (0.72 ± 3.04-90.44 ± 3.04 mg 100 g^−1^), Syringic acid (5.93 ± 1.06-22.35 ± 1.06 mg 100 g^−1^), Salvianolic acid (1.32 ± 2.62-15.86 ± 2.62 mg 100 g^−1^), Caffeic acid (5.19 ± 2.51-21.23 ± 2.51 mg/100 g^−1^), Vanillic acid (0.80 ± 0.35-4.57 ± 0.50 mg/100 g^−1^)	Croatia	[48]

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
