# Peer review of "Bioactive Profile of Various Salvia officinalis L. Preparations"

_plants, 2019, doi:10.3390/plants8030055_

Round 1

Reviewer 1 Report

Dear Authors,

this manuscript is an interesting review that describes bioactive profile of various sage preparations depending on the extraction techniques used and extraction parameters and lists the newest research findings on its health effects.

The work is well presented, but in my opinion, it needs some major revisions before publication.

Major issues:

Authors have to describe the methodology that they followed for literature selection and revision: employed databases, time interval selected, key words and so on.

As authors focus their attention on the extraction of chemical compounds very sensitive to oxidation, it could be interesting to take in account, when possible, the composition of gaseous atmosphere adopted during the extraction procedure. In particular: in which study an inert atmosphere has been adopted during extraction? This variable does can have some influence in the extract composition? In our experience the presence of O2 in extraction atmosphere deeply reduce the extraction yield of compounds sensitive to oxidation.

It could be interesting to provide an analysis of costs and benefits of the different extraction methods analyzed to better explore their effectiveness also at an industrial application level.

Other minor issues to work out:

Section 2.5: authors can implement the bibliography of this section with two interesting papers about the Sc-CO2 oil extraction from different matrices:

Venturi et al. (2017) “A simplified method to estimate Sc-CO2 extraction of bioactive compounds from different matrices: Chili pepper vs. tomato by-products” Applied Sciences (Switzerland) 7(4) art. N. 361, https://doi.org/10.3390/app7040361.

Zinnai et al. (2016) “Supercritical fluid extraction from microalgae with high content of LC-PUFAs. A case of study: Sc-CO2 oil extraction from Schizochytrium sp” J. Supercritical Fluids 116, 126-131; https://doi.org/10.1016/j.supflu.2016.05.011.

Author Response

Dear Authors,

this manuscript is an interesting review that describes bioactive profile of various sage preparations depending on the extraction techniques used and extraction parameters and lists the newest research findings on its health effects.

The work is well presented, but in my opinion, it needs some major revisions before publication.

Major issues:

Authors have to describe the methodology that they followed for literature selection and revision: employed databases, time interval selected, key words and so on.

Reply. We appreciate the reviewer's comment and the methodology for literature selection and revision has been added to the review paper.

As authors focus their attention on the extraction of chemical compounds very sensitive to oxidation, it could be interesting to take in account, when possible, the composition of gaseous atmosphere adopted during the extraction procedure. In particular: in which study an inert atmosphere has been adopted during extraction? This variable does can have some influence in the extract composition? In our experience the presence of O2 in extraction atmosphere deeply reduce the extraction yield of compounds sensitive to oxidation.

Reply. We appreciate the reviewer's comment. In the papers covered in this review, apart from paper Farhat et al. (2013), there is no mention of the work performed in an inert atmosphere. All other extractions were carried out at atmospheric pressure, with no application of the inert atmosphere.

It could be interesting to provide an analysis of costs and benefits of the different extraction methods analyzed to better explore their effectiveness also at an industrial application level.

Reply. We appreciate the reviewer's comment and an analysis of all used  extraction methods has been added to the review paper (new chapter 4)

Other minor issues to work out:

Section 2.5: authors can implement the bibliography of this section with two interesting papers about the Sc-CO2 oil extraction from different matrices:

Venturi et al. (2017) “A simplified method to estimate Sc-CO2 extraction of bioactive compounds from different matrices: Chili pepper vs. tomato by-products” Applied Sciences (Switzerland) 7(4) art. N. 361, https://doi.org/10.3390/app7040361.

Zinnai et al. (2016) “Supercritical fluid extraction from microalgae with high content of LC-PUFAs. A case of study: Sc-CO2 oil extraction from Schizochytrium sp” J. Supercritical Fluids 116, 126-131; https://doi.org/10.1016/j.supflu.2016.05.011.

Reply. These references have been read and included in the review article.

Reviewer 2 Report

Despite authors' efforts to write this paper, more work is needed to increase quality and accuracy of  this review.All the document needs to be checked by a native English reviewer. Along the paper there are many poorly constructed sentences and wrong punctuation. Also, many paragraphs are long and confusing. There are also some imprecisions regarding scientific terms and language. I recommend  authors  to improve  the document before publication

Author Response

Despite authors' efforts to write this paper, more work is needed to increase quality and accuracy of  this review.All the document needs to be checked by a native English reviewer. Along the paper there are many poorly constructed sentences and wrong punctuation. Also, many paragraphs are long and confusing. There are also some imprecisions regarding scientific terms and language. I recommend  authors  to improve  the document before publication

Reply. Our paper was extensively proofread and corrected considering the english language, sentence construction and all other mistakes. All changes are indicated within the manuscript.

Round 2

Reviewer 1 Report

Dear Author,

the manuscript has been improved and in my opinion it can be accepted in present form.

Author Response

Dear editor,

Hereby we are sending our manuscript „Bioactive profile of various Salvia officinalis L. preparationscorrected according to reviewers' comments. All corrections in the manuscript are done using Track changes.

Corrections are made according to the comments, as follows:

Rewiever 1

Dear Author,

the manuscript has been improved and in my opinion it can be accepted in present form.

Reply. Thank you very much for your response.

Reviewer 2 Report

On the attached revised version authors are requested to check again and  carefully all the manuscript regarding:

i)English language

ii)Homogeneity of compounds´ names in all  tables

iii)Correct correspondence of each reference along the manuscript.

Author Response

 Dear editor,

Hereby we are sending our manuscript „Bioactive profile of various Salvia officinalis L. preparationscorrected according to reviewers' comments. All corrections in the manuscript are done using Track changes.

Corrections are made according to the comments, as follows:

Rewiever 2

On the attached revised version authors are requested to check again and  carefully all the manuscript regarding:

i) English language

Reply. Checked as advised

ii) Homogeneity of compounds´ names in all  tables

Reply. Revised as suggested

iii) Correct correspondence of each reference along the manuscript.

Reply. Revised as suggested

The manuscript is also revised according to PDF comments:
